# ^1^H HRMAS NMR Metabolomics for the Characterization and Monitoring of Ripening in Pressed-Curd Ewe’s Milk Cheeses Produced Through Enzymatic Coagulation

**DOI:** 10.3390/foods14132355

**Published:** 2025-07-02

**Authors:** David Castejón, José Segura, Karen P. Cruz-Díaz, María Dolores Romero-de-Ávila, María Encarnación Fernández-Valle, Víctor Remiro, Palmira Villa-Valverde, María Isabel Cambero

**Affiliations:** 1ICTS Complutense Bioimaging (BioImac), Complutense University of Madrid, Paseo Juan XXIII, 1, 28040 Madrid, Spain; dcastejon@ucm.es (D.C.); evalle@ucm.es (M.E.F.-V.); palmira@ucm.es (P.V.-V.); 2Department of Food Technology, Faculty of Veterinary Medicine, Complutense University of Madrid, Av. Puerta de Hierro s/n, 28040 Madrid, Spain; kcruzdiaz@gmail.com (K.P.C.-D.); mdavilah@ucm.es (M.D.R.-d.-Á.); vremiro@ucm.es (V.R.); icambero@ucm.es (M.I.C.)

**Keywords:** ^1^H HRMAS NMR, *Manchego* cheese, *Castellano* cheese, artisanal, industrial, chemometrics, ripening, metabolic profile

## Abstract

A comprehensive characterization of two pressed-curd cheeses produced from ewe’s milk using enzymatic coagulation—*Manchego* cheese (with Protected Designation of Origin, PDO) and *Castellano* cheese (with Protected Geographical Indication, PGI)—was performed throughout the manufacturing process (industrial or traditional) and ripening stages (2, 9, 30, 90, and 180 days). Proton high-resolution magic angle spinning nuclear magnetic resonance (^1^H HRMAS NMR) spectroscopy, combined with Principal Component Analysis (PCA) and cluster analysis, was applied to intact cheese samples. The combination of this spectroscopic technique with chemometric methods allows for the characterization of each type of sheep milk cheese according to its geographical origin and production method (artisanal or industrial), as well as the estimation of ripening time. The results demonstrate that HRMAS NMR spectroscopy enables the rapid and direct analysis of cheese samples, providing a comprehensive profile of their metabolites—a metabolic ‘fingerprint’.

## 1. Introduction

Southern European countries are among the leading producers of hard and semi-hard cheeses made from ewe’s milk. Many of these cheeses are traditionally manufactured using animal rennet as the coagulant and highly valued for their distinctive sensory attributes. Their characteristic aromas and flavors, which develop during the ripening process, are primarily the result of the enzymatic degradation of milk fats, proteins, and carbohydrates [1]. A wide variety of ewe’s milk cheeses (EMC) are produced throughout the Mediterranean basin, with Spain standing out due to its diverse range of traditional pressed-curd cheeses made from ewe’s milk using enzymatic coagulation, such as *Manchego* (Protected Designation of Origin, PDO, Regulation (EC) 1107/96) [2] and *Castellano* cheese (Protected Geographical Indication, PGI, Regulation (EU) 2020/247) [3].

*Manchego* is produced in the central Spanish region of Castilla–La Mancha (CLM cheese) and internationally recognized and extensively studied [4,5]. It is made exclusively from the milk of Manchega sheep. In contrast, *Castellano* cheese is produced in the northwestern region of Castilla y León (CL cheese) using milk from Churra and Castellana sheep breeds. This cheese is characterized by a firm texture and a mild flavor, and although it is well-appreciated by consumers, it remains less well-known in the scientific literature. To date, only a limited number of studies have addressed its microbiota, volatile compound profile, texture, and other sensory characteristics [5,6,7,8]. Both CLM and CL are semi-hard, with similar external appearances and closely related manufacturing technologies. Typically, their production involves enzymatic coagulation and, in most cases, the addition of mesophilic starter cultures, commonly strains of *Lactococcus lactis* ssp. *lactis* and *Lactococcus lactis* ssp. *Cremoris*. The curd is then pressed and, eventually, subjected to a ripening period [9]. These cheeses are produced either traditionally (artisanal) or industrially. Traditional (T) cheeses are made from raw milk in local cheesemaking facilities, while industrial (I) cheeses are produced at a factory scale using either raw or pasteurized milk under strictly controlled conditions. In general, bovine rennet is used for CLM, whereas ovine rennet is preferred in CL. The commercial value of these cheeses is directly linked to their organoleptic quality, which depends on both the ripening time (RT) and the production process. Therefore, a detailed characterization of their chemical properties at different RTs and under varying production conditions would be particularly valuable for ensuring product authenticity and preventing fraudulent practices [10].

Nuclear magnetic resonance (NMR) spectroscopy, particularly proton NMR (^1^H NMR), has gained increasing attention as a rapid and accurate analytical technique for the investigation of biological matrices [11,12]. Among its various modalities, high-resolution magic angle spinning (^1^H HRMAS NMR) has proven especially valuable for the direct analysis of intact semi-solid samples, providing high-resolution spectra without the need for extensive sample preparation [13]. Unlike other analytical methods, ^1^H HRMAS NMR is minimally invasive, requiring only small sample quantities (approximately 15 mg), and it enables the simultaneous detection and quantification of a broad range of metabolites, both polar and apolar, in a single experiment [14]. This eliminates the need for extraction or separation procedures, thereby minimizing the risk of sample alteration, loss, or contamination. Furthermore, the technique aligns with current environmental standards by reducing the use of solvents and reagents, contributing to more sustainable laboratory practices. Due to these advantages, ^1^H HRMAS NMR has emerged as a powerful tool for the characterization of semi-solid foods and biological tissues, offering detailed metabolic profiles. Over the past decade, it has been successfully applied to a wide range of food matrices, including various cheese types, such as *Mozzarella* [15], *Fiore Sardo* [16], and PGI *Castellano* cheese [8], as well as cereals, fruits, vegetables, dairy products, and fish and meat [17,18].

Metabolomics is a comprehensive analytical approach within systems biology that aims to identify and quantify the complete set of low-molecular-weight metabolites in a biological sample. It integrates advanced analytical techniques, primarily NMR spectroscopy and mass spectrometry, with multivariate statistical and chemometric tools to interpret complex datasets. These techniques enable the detection of subtle biochemical changes associated with physiological states, environmental influences, or technological processes. In the context of food science and technology, metabolomics has emerged as a powerful tool for characterizing the chemical composition of food products, assessing quality and authenticity, monitoring fermentation and ripening processes, and evaluating the impact of processing and storage [19,20,21]. NMR-based metabolomics offers the advantages of NMR spectroscopy, enabling minimally invasive analysis with high reproducibility, making it especially suitable for the study of complex food matrices.

A variety of chemometric methods are available to extract meaningful information from complex and high-dimensional datasets generated through NMR spectroscopy. These methods are commonly categorized according to the types of analytical problems they address [22]. First, there are techniques aimed at simplifying large and intricate datasets and uncovering hidden patterns within them, such as Principal Component Analysis (PCA). Second, classification and discrimination methods are employed, which can be further divided into unsupervised and supervised approaches. The latter require a training set of samples with known class membership. Examples of supervised methods include Linear Discriminant Analysis (LDA) and Soft Independent Modeling of Class Analogy (SIMCA). For quantitative analysis, particularly in cases involving severe signal overlap or when analyte signals cannot be directly observed, methods like Principal Component Regression (PCR), Partial Least Squares (PLS) regression, Multivariate Curve Resolution (MCR), and Independent Component Analysis (ICA) are commonly applied [22].

In this context, the combination of ^1^H HRMAS NMR spectroscopy with multivariate chemometric methods represents a highly effective approach, as it enables the acquisition of a comprehensive metabolic fingerprint from an intact semi-solid sample in a single experiment [10,23]. Consequently, any factors that induce variations in this metabolic profile can be specifically targeted for investigation. Indeed, ^1^H HRMAS NMR spectroscopy, when applied within a metabolomics framework, has been recognized as a reliable tool for assessing various quality-related attributes of agri-food products, including nutritional value, geographical origin, authenticity, genotype, and safety. With this objective, ^1^H HRMAS NMR–metabolomic has been successfully applied to the analysis of diverse matrices, such as beef and pork [17,24,25], game meat [26], different fruits [14], and several types of cheese, among other products. Galli [27] emphasized that recent advancements in omics sciences have become essential for the characterization and authentication of PDO cheeses, as they allow for the identification of unique sensory, chemical, and biological features associated with specific micro-regions. This scientific approach not only supports the authentication of these products but also contributes to broader objectives, such as biodiversity preservation and sustainable production practices.

In this study, ^1^H HRMAS NMR spectroscopy was employed to monitor metabolomic evolution throughout the ripening period of two pressed-curd cheeses made from ewe’s milk using enzymatic coagulation produced in Spain: PDO *Manchego* and PGI *Castellano* cheeses. The analysis encompassed both traditional cheeses (produced from raw milk) and industrial cheeses (produced from pasteurized milk). The primary objective of this work was to assess the suitability of combining this analytical technique with multivariate statistical analysis, particularly Principal Component Analysis (PCA), to characterize and classify the cheeses according to their ripening stage and manufacturing process. This research builds on previous studies [8] in which acquisition parameters were optimized to obtain high-quality spectra and standard analytical conditions were established for the analysis of this type of cheese. Both *Manchego* and *Castellano* cheeses are considered representative models of pressed-curd EMC produced through enzymatic coagulation. The findings obtained in this study may serve as a foundation for future research on this category of cheeses or similar dairy products.

## 2. Materials and Methods

### 2.1. Experimental Design and Sampling

The ewe’s milk cheeses (EMC), referred to as CLM and CL throughout this study, were produced following the procedures described in the EU regulations for PDO *Manchego* cheese [2] and PGI *Castellano* cheese [3], respectively. The I-CLM and I-CL samples were produced in an industrial facility, whereas the T-CLM and T-CL samples were elaborated in small-scale cheeseries following a traditional/artisanal procedure, using commercial calf dried rennet and commercial natural lamb rennet, respectively. T-CLM, I-CL, and T-CL cheeses were manufactured from raw milk. The specific features of the production process are summarized in Appendix A. To produce all cheeses, molds with a diameter of 19–20.5 cm and a height of 14–15.5 cm were used. The weight of the fresh cheeses ranged from 3.7 to 4.3 kg, decreasing to 3.0 and 3.5 kg by the end of the ripening period.

For the NMR studies, 2, 9, 30, 90, and 180 days of ripening time (RT) were considered. Five pieces of each type of cheese were analyzed [two cheese varieties (CL and CLM) × two processes of elaboration (I and T) × five RT × five cheese samples].

For each of them, we excluded the rind (1 cm from the outside) and the central (1 cm) section. Three core samples were subsequently collected from each cheese using a biopsy punch (Stiefel, 3 mm diameter). Each analysis was carried out in triplicate.

### 2.2. ^1^H HRMAS NMR Analyses

Samples were stored at 4 °C until analysis. In a 50 µL HRMAS rotor insert, 20 µL of 3-(trimethylsilyl) propionic-2,2,3,3-d_4_ acid sodium salt (TSP) solution in D_2_O 1 mM (TSP and D_2_O, Scharlau, Spain) was added together with 10 ± 2 mg of sample. The inserts were placed in a Zirconium rotor, and ^1^H HRMAS NMR spectroscopy was carried out at 25 °C with a 500 MHz spectrometer Bruker AMX500 (Bruker GmBH, Rheinstetten, Germany). All experiments were conducted with a spinning speed of 6000 Hz, following the protocol optimized for EMC by Castejón et al. [8]. One-dimensional (1D) NOESY experiments with pre-saturation were performed using a recycling delay of 1 s and an acquisition time of 2.25 s. A spectral width of 8333 Hz was employed. 1D CPMG experiments were performed using two different echo times, 50 and 400 ms, and a recycling delay of 3 s with an acquisition time of 1.57 s. A spectral width of 10,417 Hz was employed. For all experiments, 128 scans were collected into 32 K data points. Metabolite identification was confirmed with the aid of 2D homonuclear NMR experiments, including ^1^H,^1^H-COSY-HRMAS (correlated spectroscopy) and ^1^H,^1^H-TOCSY-HRMAS (total correlated spectroscopy). The spectra were processed with zero-filling prior to Fourier transformation [8].

### 2.3. Data Treatment and Statistics

Assignment of resonances in the ^1^H HRMAS NMR spectra from cheese was based on both spin connectivity information obtained from 2D experiments and the use, as guidelines, of both data reported in the literature [8,28,29] and data obtained from the HMDB database (Human Metabolite Data Base: http://www.hmdb.ca) (last accessed on 17 May 2025). Baseline correction was performed on all one- and bi-dimensional spectra. The frequency axes were calibrated by referencing the ^1^H signals of TSP to 0 ppm. D_2_O was employed to provide the lock signal, thereby ensuring the stability of the spectrometer’s magnetic field. All spectra were processed by using both Bruker Topspin Software (v.4.3) and MestReC NMR Processing Software (v.4.9.9.9).

Statistical analyses were carried out using Statgraphics Centurion XVI for Windows to determine the effects of RT and the elaboration process. Duncan’s test for the multiple mean comparisons procedure was used (95% confidence level). Mean values and their standard deviation were used for data presentation. Multivariate statistical analysis was conducted on NMR data. To identify metabolite differences associated with the cheese variety and the ripening time, Principal Component Analysis (PCA) and cluster analysis were performed. PCA was carried out using Statgraphics Centurion XVI for Windows. The number of principal components (PCs) employed for PCA was established as the minimum required to explain 95% of the total variance. Additionally, AMIX software (version 3.9.11, Bruker BioSpin, Ettlingen, Germany) was used. Prior to the multivariate analyses, each individual spectrum was data-reduced over the 9.03–0.76 ppm range by dividing it into spectral regions (buckets) of variable widths. These variable widths were chosen to account for the chemical shift variations due to working with non-buffered samples. Regions with spectral artifacts (i.e., unsuppressed water regions) were excluded from the bucketing. In total, 83 buckets were defined, and the integral of each bucket was normalized to the sum of all of the integrals of the spectrum. The variable buckets used for the multivariate analysis, including their midpoint and their width, are provided in Appendix A. To account for the variable concentration of each reconstituted sample, bucket intensities were normalized to the total spectral intensity.

For the unsupervised PCA, a 161 × 18 matrix was constructed, with its rows representing the different cheese samples (cases) and its columns representing the integrated buckets (variables). These columns were scaled to unit variance prior to PCA calculations, resulting in all of the buckets becoming equally important; in short, case clustering, if evidenced, should be explained by differences in the metabolite profile of the cases rather than in their metabolite absolute levels. Significance analysis of variables (buckets) was based on the procedure of Goodpaster et al. [30] using a confidence level of 95%. Cluster analysis was conducted using the methodology implemented in the SPAD software package (SPAD.N, 2003), ‘Système Portable pour l’Analyse des Données’ (Version 5.6, CISIA, Montreuil Cedex, France).

## 3. Results and Discussion

### 3.1. Evolution of Metabolites During the Ripening of Pressed-Curd Cheeses Made from Ewe’s Milk Using Enzymatic Coagulation: I-CLM, T-CLM, I-CL, and T-CL

In a previous study [8] conducted with I-CL and T-CL cheeses, including the ewe’s milk used for their production, the main acquisition parameters, such as temperature, echo time, spinning rate, gradient strength, and spectral width, were optimized to obtain ^1^H HRMAS NMR spectra suitable for pressed-curd EMC using enzymatic coagulation. In the present work, the metabolites of EMC with very similar characteristics, I-CLM and T-CLM, were analyzed through ^1^H HRMAS NMR spectroscopy. This study provides a comprehensive analysis of the ripening process of CLM cheeses (from both I and T production methods). Furthermore, CL cheeses (from both I and T processes) were also studied to complement the data reported by Castejón et al. [8], who previously described their metabolome. The findings from all four cheese varieties converge in a final analysis that includes PCA and cluster analysis, aimed at evaluating the classification potential of ^1^H HRMAS NMR–metabolomic for cheeses differing in origin, production method, and RT.

Figure 1 presents representative ^1^H HRMAS NMR spectra obtained during the ripening process (at 2, 9, 30, 90, and 180 days) for I-CL and I-CLM cheeses. These spectra provide information on a wide range of compounds, both major and minor, including amino acids, fatty acids, lactose derivatives, organic acids, cholesterol, phospholipids, and triglycerides, among others. Similar spectra were obtained with traditionally made cheeses (Appendix A). A total of 116 signals corresponding to different metabolites were identified in four varieties of EMC (CLM and CL from production I and T) (Appendix A). Several of these metabolites are associated with the development of flavor and aroma, suggesting that this spectroscopic approach could serve as a rapid tool for estimating cheese sensory quality and monitoring the biochemical processes of glycolysis, proteolysis, and lipolysis involved in cheese ripening. Subsequently, changes in various metabolites detected in the ^1^H HRMAS NMR spectra were analyzed.

#### 3.1.1. Carbohydrates

In the spectra from I-CL cheeses (Figure 1) at 2, 9, 30, and 90 days of RT, a group of signals corresponding to lactose (3.5–4.0 ppm) and its constituent monosaccharides (α/β-galactose and α/β-glucose), along with a β-galactose doublet at 4.58 ppm, were observed. However, these signals were no longer detectable at 180 days. Similar patterns were observed in the spectra of T-CLM cheeses (Appendix A). These results are consistent with those reported by several authors [31,32], and the same monosaccharides have been identified through NMR analysis in other types of cheese, such as in *Mozzarella* [15] and *Fiore Sardo* [16]. The evolution of these metabolites during ripening is attributed to the metabolism of residual lactose, which is rapidly converted into its constituent monosaccharides through the action of β-galactosidase, primarily produced by homofermentative lactic acid bacteria (LAB) starter cultures. Notably, I-CLM cheeses (Figure 1) showed no detectable signals for lactose or galactose at any of the analyzed time points. These differences between different varieties of cheese may be related to the particularities of their manufacturing process. Several authors have related a similar decrease in the intensity of these signals at the beginning stages of the maturation process to lactose metabolism, but they have also mentioned that such a difference could also be related to the fact that, after coagulation and cutting, washing of the curd is usually carried out in the industry, thus replacing whey, containing lactose in solution, with water. With this procedure, both lactate formation and the pH value would be controlled to ensure homogeneity and avoid the production of too much acid, sandy or thick cheese, or cheese with a strong taste [33,34].

#### 3.1.2. Fatty Acids

Conjugated linoleic (5.90 ppm), caproleic (5.76 ppm), linoleic (2.85 ppm), linolenic (2.78 ppm), and butyric acids (0.94 ppm) were detected in the ^1^H HRMAS NMR spectra (Appendix A). The intensity of the signal associated with butyric acid showed statistically significant changes (*p* < 0.05) related to RT and the elaboration process for all of the considered cheese varieties (I-CL and T-CL and I-CLM and T-CLM). Greater signal intensity was observed in I-CLM than in T-CLM cheeses until day 30 of analysis and no difference was observed at day 90, while the analysis at day 180 showed a greater intensity for T-CLM than for I-CLM. Similarly to CLM, I-CL values were higher than those of T-CL at the early ripening stages. However, an opposite trend was observed with RT between I-CL and T-CL, with I-CL showing a 4.9% decrease and T-CL exhibiting a 4.1% increase (Table 1; *p* < 0.05). Buffa et al. [35] detected lower lipolysis in cheese from pasteurized milk because thermal treatment inactivated lipoprotein–lipase. Therefore, the lower amount of butyric acid in cheeses manufactured with pasteurized milk (I-CLM) at day 180 of ripening compared to those manufactured with raw milk (T-CLM) could be explained by the partial denaturalization of the native lipases of milk.

The associated signals for CLA showed no significant differences (*p* > 0.05) between types of elaboration for CLM (I-CLM and T-CLM), but T-CL values were 29.7% higher than those of I-CL. Moreover, no global change was detected between days 2 and 180 of ripening for CL, resulting in an increase of a small magnitude for CLM (around 4%, Table 2). In line with our results, Jang et al. [36] and Govari et al. [37] found that CLA content is highly dependent on the amount in the original milk and little dependent on the starter culture and ripening. Both studies demonstrated that specific probiotic LAB used as the starter culture can enhance CLA levels in cheese during ripening and that seasonal variations and animal diet significantly influence the CLA content in raw milk.

Linoleic acid showed higher signal intensity in I-CL than in T-CL (65.8% difference on average; *p* < 0.001, Table 1). No significant change with RT was detected for linoleic acid I-CL, and a 60% increase was detected for T-CL between day 2 and day 180 values. However, only statistical tendencies were observed for higher I-CLM signal intensities compared to T-CLM and for an increase with RT. Regarding linolenic acid, only statistical tendencies for T-CL values were higher than those of I-CL, and a tendency to increase during RT was detected (*p* = 0.0609 and *p* = 0.0601, respectively).

Free fatty acids are the result of lipolysis during ripening, which is related to the lipase and esterase activity of native microbial enzymes from milk and rennet. Free fatty acids also contribute to the taste and aroma [9]. Lower values of 2% have been described for lipolytic processes on cheese triglycerides [9,29]. On one hand, different lipolytic behaviors could be a consequence of thermal pre-treatment in I- but not in T-cheeses, thus implying an increase in the hydrolysis rate due to the temperature increase followed by faster enzyme inactivation. Moreover, these results could be related to the positional distribution within the triglyceride molecule, with short-chain fatty acids being mainly located in the sn-1 and/or the sn-3 position, together with the activity, even at low concentrations, of the native lipoprotein–lipase of milk [38]. Thus, triglyceride sn-1 and/or sn-3 location hydrolysis is both kinetically and thermodynamically facilitated.

#### 3.1.3. Organic Acids

The profiles of lactic (4.15 ppm), citric (2.71 ppm), acetic (1.95 ppm), and formic (8.45 ppm) acids along RT are shown in Figure 2.

Regarding lactic acid, I-CL and I-CLM values were 12.1% and 28.2% higher than T-CL and T-CLM values, respectively, at day 2 of RT (*p* < 0.0001). Such a difference decreased to 4.1% and 8.6% on day 9, and no differences were detected after day 9 (Figure 2A).

Regarding acetic acid, on day 2 of analysis, I-CL presented a lower relative signal intensity than T-CL (31.5%), but the T-CLM value was 37.3% higher than the I-CLM one at day 30. Remarkably, T-CL and T-CLM values on day 30 of ripening were 50.5% and 39.7% higher than day 9 values, but no statistically significant differences were detected for I-CL and I-CLM (Figure 2B). Such a difference between elaboration processes was observed for the rest of the ripening.

A greater decrease in the citric acid amount was observed between days 2 and 30 of analysis in T-CLM compared to I-CLM (39.5% and 10.4%, respectively; *p* < 0.05 for type × time interaction), possibly explained by lower activity of the microbiota in I-CLM. An opposite response was detected for the levels of citric acid with RT between elaboration processes. A decrease of a small magnitude in signal intensity was detected for I-CL, whereas an increase was observed for T-CL (*p* < 0.05; Figure 2C).

Considering formic acid, an increase in the signal intensity was detected in CL varieties. Nevertheless, I-CL presented greater formic acid relative signal intensity on day 2 of processing compared to T-CL (*p* < 0.001), but, on days 90 and 180 of ripening, no significant differences were found. For the CLM variety, no statistically significant differences were detected between both types of cheese for days 2, 9, and 30 of analysis. Nevertheless, a higher decrease in T-CLM than in I-CLM was found in the last 5 months of ripening (71.4% and 48.0%, respectively; *p* = 0.048 for type × time interaction) (Figure 2D).

The different compositional changes of lactic, acetic, formic, and citric acids observed during the ripening period have been attributed to the metabolic activity associated with both the initial and residual lactose. These metabolic processes are closely linked to several factors, including the composition of the starter culture, the presence of active native microbiota (not inactivated by pasteurization), the specific composition of the rennet, and the potential application of curd washing [39].

A similar RT evolution to our results was detected for lactic acid with other types of cheese. Differences related to the elaboration process may be associated with the fact that the lactose metabolic pathway in cheese production depends on the presence of different LAB. Homofermentative LAB (*Streptococcus*, *Enterococcus*, *Lactococcus*, *Lactobacillus*, etc.) metabolize most of the glucose content into lactic acid. In addition, heterofermentative LAB (*Leuconostoc* and some *Lactobacillus*) produce lactate, ethanol, acetate, and CO_2_ when metabolizing glucose. The increase in the relative intensity of the formic acid signal could be associated with the presence of *Streptococcus thermophilus*, which produces formic acid, pyruvic acid, and CO_2_ from lactose. Lombardi et al. [40] for *Reggianito* and Califano and Bevilacqua [41,42] for *Gouda* and *Mozzarella* cheeses observed similar behavior in these maturation times.

Similar performance of acetic acid was previously observed in *Cheddar* cheese by Lues and Bekker [43] and in *Reggianito Argentino* cheese [44]. Acetic acid can be produced through various pathways, such as the fermentation of lactose by heterofermentative LAB, the breakdown of amino acids like alanine, glycine, aspartic acid, and serine by LAB, and through the metabolic processing of citrate [38].

Attending to citric acid, Lombardi et al. [40] and Ballesteros et al. [45] observed similar behavior in *Reggianito* and *Manchego* types of cheese, respectively. Because citrate is a subtract of the Krebs cycle, it is usually transformed into pyruvate, α-acetolactate and diacetyl, CO_2_, acetoin, and 2,3-butylene glycol by the citrate-positive strains of the present microbiota, such as Lactococcus lactis, some species of *Leuconostoc*, and/or other non-starter lactic acid bacteria (NSLAB), such as *Lactobacillus plantarum* [46,47]. Industrial production is focused on the control of the native microbiota to obtain cheeses with uniform quality. Then, the NSLAB are especially important in T-CLM as part of the native microbiota, which is altered by the thermal treatment of milk in I-CLM, as previously described.

#### 3.1.4. Amino Acids

Table 2 shows the evolution of the detected amino acids along RT for two elaboration procedures in both types of cheese, *Manchego* (I-CLM and T-CLM) and *Castellano* (I-CL and T-CL). Relative intensity values of all amino acids increased with RT. However, the change depended on the elaboration process. The lowest increase values were detected for I-CL Met (19.5%), Gln (24.7%), Ile (18.0%), and Leu (26.4%). At the end of ripening, T-CL showed higher values than I-CL. For CLM, similar behavior was detected, except for Trp, Thr, Ile, Leu, and Pro.

Most of the literature has related an increase in the total amount of free amino acids at the end of ripening, thus implying proteolytic processes during ripening, which were previously related to the taste of aged cheeses [48,49]. The main proteolytic enzymes include milk native proteases, milk-coagulating enzymes preserved in curd, and peptidases and proteases released from starter and non-starter cultures. Conversely, free amino acids are mostly released by starter culture enzymes’ activity. A different profile has been described depending on the use of pasteurized or raw milk. Thermal treatment has been reported to alter the protein’s (mainly casein) conformation and interaction, thus hampering the accessibility of any protease to the substrate. In addition, access to hydrophobic sections of the peptides would be easier, hence the higher amount of hydrophobic amino acids compared to hydrophilic ones [50,51]. Moreover, thermal treatment could also alter the activity of exogenous microbiota. Microbial proteases and peptidases would probably also be inhibited [52,53]. In addition, the more complex microbiota from cheeses manufactured with raw milk was related to quicker maturation and a higher degree of proteolysis than those made from pasteurized milk, thus contributing to a more intense aroma and flavor [48].

### 3.2. Potential of ^1^H HRMAS NMR for the Characterization and Monitoring of the Ripening Process of Pressed-Curd Cheeses Made from Ewe’s Milk Using Enzymatic Coagulation

To assess the potential of the data obtained from the ^1^H HRMAS NMR analysis for the characterization of the studied EMC and the monitoring of their ripening process, unsupervised multivariate statistical methods, PCA and cluster analysis, were employed. These methods allow for the evaluation of data variability without considering predefined group classifications, and they are highly useful for identifying patterns and/or trends within the studied samples [54].

Initially, unsupervised PCA was employed as a criterion for the complete metabolome association, with the aim of enabling EMC grouping according to RT and the production method (industrial vs. traditional). This analysis began with the spectral data from CLM cheeses. For this purpose, each spectrum was segmented into spectral regions, or buckets, of variable widths. A total of 83 buckets (variables) × 50 spectra were considered in the analysis. Given that these buckets include metabolites with high signal intensity (such as fatty acids), as well as others with much lower intensity (such as lactose), various transformation procedures were applied to homogenize and mitigate the effect of intensity differences. The best results were obtained using unit variance scaling [55]. The midpoint and width of the buckets considered in the analysis, along with the associated metabolites and the loading coefficients for each factor in the first two principal components (PC1 and PC2), are presented in Appendix A.

This procedure enabled the differentiation of CLM cheeses according to the production method (I-CLM vs. T-CLM) and RT (Figure 3A). In total, 96.18% of the variance was explained by ten PCs, although the variance explained by the first two principal components was limited to 66% (46.13% and 19.83% for PC1 and PC2, respectively). To increase this percentage, a new PCA was performed using only the buckets corresponding to amino acids, as these metabolites exhibited the greatest variability among the cheeses studied. No scaling was required for the processing of these buckets, as their signal intensities were within a similar order of magnitude. The first two components (PC1 and PC2) obtained through this approach explained 88.8% of the total variance (PC1: 78.0% variance vs. PC2: 10.83% variance) and yielded a grouping pattern similar to that obtained using the full metabolome (Figure 3B, Appendix A). A good sample pooling was observed along RT in PC1 and depending on the elaboration process (I vs. T) in PC2. The same plot (PC1 vs. PC2) with the amino acids is shown in Figure 3C. The PCA indicated that Asp, Pro, Ile, Trp and Leu had higher presence in the I-CLM cheeses; while Val, Thr, His, Met, Phe, Tyr, Glu, and Gln showed greater importance in T-CLM cheeses. These results could be related to the already mentioned thermal treatment of milk [51]. It is also worth noting the role of native LAB [38], which may contribute to the differences observed between industrially and traditionally produced cheeses. Additional factors that may influence this differentiation include the use of various types of natural rennet or recombinant chymosin (Appendix A), as well as the different sheep breeds used for milk production [2,3]. These elements would help distinguishing cheeses from CL and CLM. This finding suggests that the combination of ^1^H HRMAS NMR spectral data for amino acids with PCA is sufficient to differentiate the CLM cheeses studied—based on both ripening time and production method—while significantly simplifying data processing.

Following this initial analysis, the results obtained from the four cheese varieties (I-CL, T-CL, I-CLM, and T-CLM) were jointly analyzed using PCA (Figure 4) based on the amino acid buckets. For the same reason mentioned previously, the spectral regions corresponding to amino acids were used in this analysis, thereby enhancing the variance explained by the first PC. Specifically, the first two PCs accounted for 89.2% of the total variance (PC1: 66.99% and PC2: 22.19%). The score plot of the analyzed cheese samples (Figure 4, Appendix A) reveals a parallel and differentiated evolution of both varieties of cheese [CL (I and T) vs. CLM (I and T)] throughout the ripening process. With PC1, the samples were distributed according to their RT, whereas PC2 differentiates them based on their geographical origin (CL vs. CLM) and/or production process (I vs. T). Thus, sheep cheeses from Castilla y León, I-CL and T-CL, make a greater contribution to PC2 and are grouped and located at the top, while CLM cheeses are grouped and located at the bottom. These results further underscore the potential of ^1^H HRMAS NMR spectroscopy to monitor the cheese ripening process, as well as for distinguishing varieties of EMC that exhibit very similar morphological characteristics but differ in the sheep breed used for milk production, the geographical origin, and various factors related to their manufacturing processes (Appendix A), all of which contribute to changes in their metabolome. One of the first studies in this line of research was conducted on PDO *Mozzarella di Bufala Campana* cheeses [15], where ^1^H HRMAS NMR spectroscopy enabled rapid characterization of the metabolic profile and classification based on both geographical origin and freshness.

A detailed analysis of the sample distribution (Figure 4) revealed a clear separation between CLM cheese samples produced using industrial methods (positioned lower) and those produced using traditional methods (positioned higher). A similar distribution pattern was observed for CL cheeses, although the separation between samples from different production processes (I-CL and T-CL) was notably smaller than that observed for CLM cheeses. The differentiation of CLM cheese samples according to production method (I-CLM vs. T-CLM) has been attributed to the use of pasteurized milk in I-CLM and raw milk in T-CLM. In contrast, the closer proximity of industrially (I-CL) and traditionally (T-CL) produced CL cheeses is likely because both were made using raw milk (Appendix A). In this case, the observed differences may be related to the native microbiota and the use of natural ovine rennet in the artisanal cheese (T-CL).

To advance the unsupervised statistical analysis of the data, a cluster analysis was conducted using metabolites that exhibited well-resolved and isolated signals, had been reliably assigned to specific chemical components, and showed the greatest contribution to sample differentiation in prior PCA models (Appendix A). Based on these criteria, 26 unscaled buckets were selected for analysis. The resulting clusters were characterized by variables whose mean values within each cluster deviated significantly from the overall mean value (Table 3). As shown in Figure 5, the samples were partitioned into three distinct clusters. This clustering approach enabled clear differentiation between cheeses originating from the two geographical regions under study. All cheese samples from CLM were assigned to Cluster 2 (Table 3), whereas cheeses from CL were distributed between Cluster 1, primarily composed of industrially produced cheeses (I-CL) with less than 180 days of ripening, and Cluster 3, which comprised traditionally produced cheeses (T-CL) aged for more than 30 days, as well as industrial cheeses (I-CL) at advanced stages of maturation. Cluster 2 (CLM cheeses) was characterized by elevated levels of butyric and acetic acids and leucine and reduced levels of glutamic acid, glutamine, methionine, and other amino acids. In contrast, Cluster 3 (traditionally produced and long-aged CL cheeses) exhibited higher concentrations of free amino acids, linoleic acid, and most other metabolites analyzed, except for butyric acid. Additionally, Figure 5 illustrates the spatial distribution of cluster centroids at various ripening stages, revealing a progression consistent with the maturation process. These findings corroborate the sample differentiation patterns previously identified through PCA.

## 4. Conclusions

^1^H HRMAS NMR spectroscopy enables the direct analysis of intact cheese samples, providing a comprehensive profile of their metabolites—a metabolic ‘fingerprint’. The combination of this spectroscopic technique with chemometric methods, such as Principal Component Analysis (PCA) and cluster analysis, can be effectively used for the characterization of pressed-curd cheeses made from ewe’s milk using enzymatic coagulation and produced in Spain, specifically PDO *Manchego* and PGI *Castellano* cheeses. Moreover, this approach allows for monitoring the ripening process. For the classification of these cheeses according to ripening time and production method (industrial vs. traditional), either the full set of identified metabolites or only the amino acids detected in the ^1^H HRMAS NMR spectra can be used.

The obtained results open numerous opportunities for the application of ^1^H HRMAS NMR spectroscopy in the quality control and traceability of high-value cheeses, such as those made from ewe’s milk. Both *Manchego* and *Castellano* cheeses are considered representative models of pressed-curd ewe’s milk cheeses produced through enzymatic coagulation. The findings obtained in this study may serve as a foundation for future research on this category of cheeses or similar dairy products.

## Figures and Tables

**Figure 1 foods-14-02355-f001:**
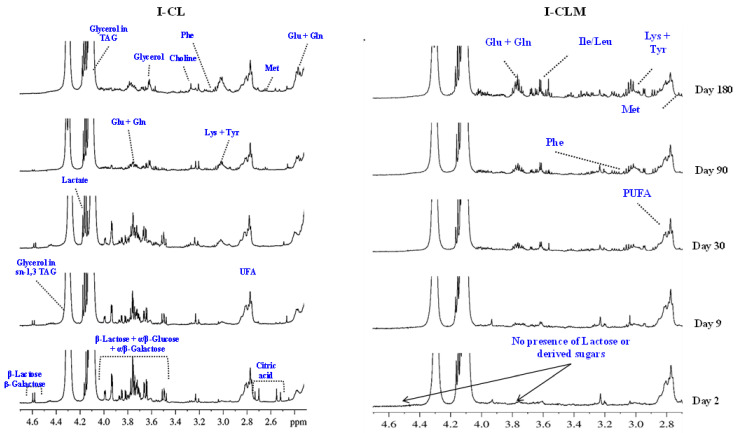
Example of ^1^H HRMAS NMR spectra (2.6–4.7 ppm) showing the evolution of different metabolites throughout the ripening time (2, 9, 30, 90, and 180 days) of industrially processed cheeses of Castilla y León (I-CL) and Castilla–La Mancha (I-CLM).

**Figure 2 foods-14-02355-f002:**
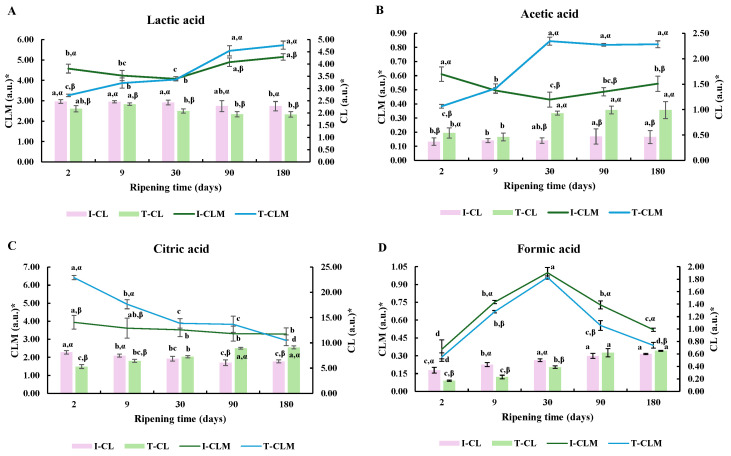
Relative signal intensity of lactic acid (**A**), acetic acid (**B**), citic acid (**C**) and formic acid (**D**) from ^1^H HRMAS NMR spectra. * Relative signal intensity (arbitrary units, a.u.) corresponding to the ratio of the NMR signal integral to the sum of all NMR signal integrals observed. I- and T- = industrial and traditional/artisanal procedures according to Appendix A. Different letters a, b … indicate statistically significant differences (*p* < 0.05) regarding RT. Different letters α, β … for each compound indicate statistically significant differences (*p* < 0.05) regarding the manufacturing process (T vs. I).

**Figure 3 foods-14-02355-f003:**
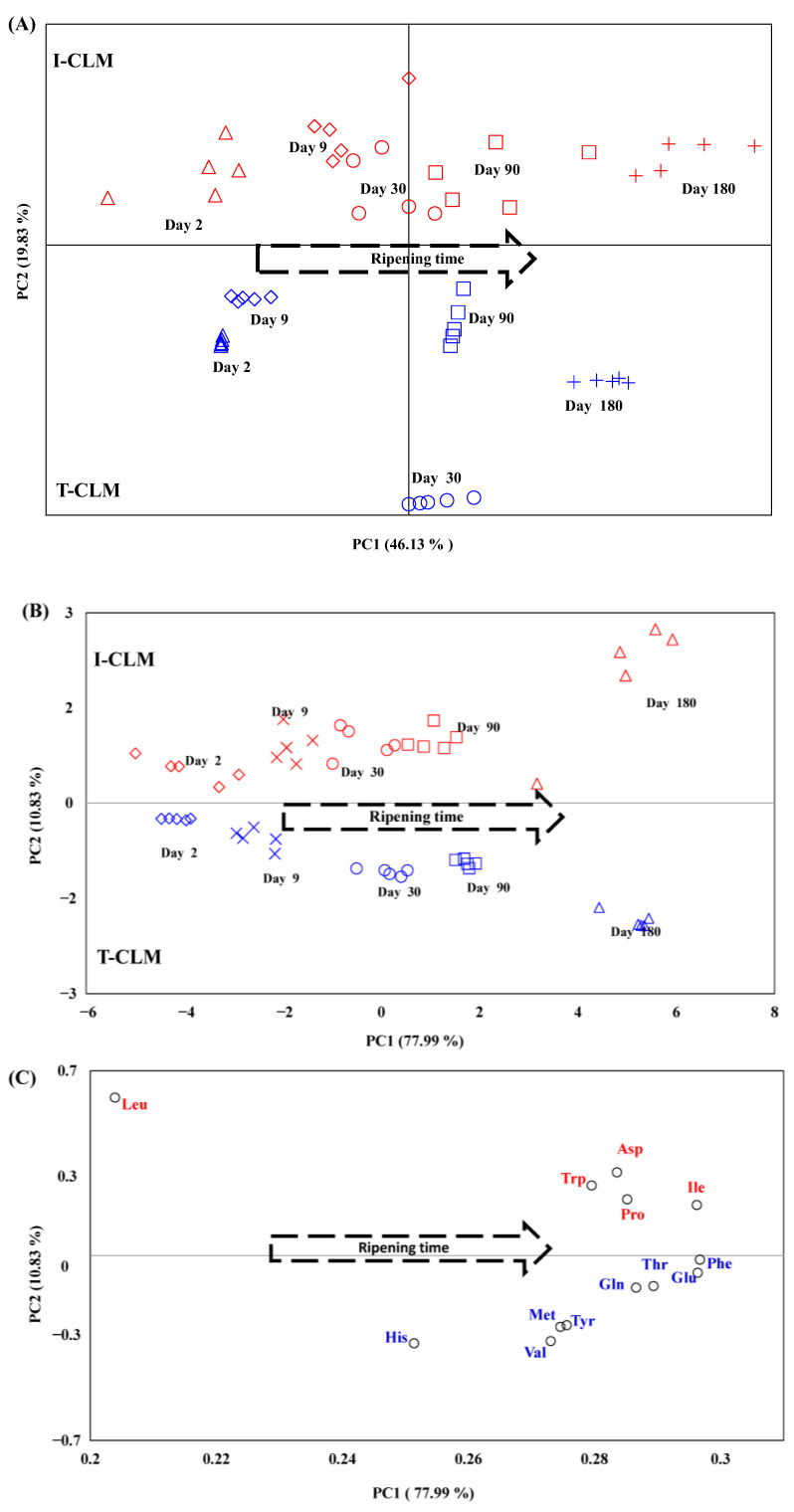
Distribution of *Manchego* cheese (CLM) samples produced through industrial (I) and traditional (T) methods at different ripening times (days), projected onto the first two principal components (PC1 and PC2) obtained from unsupervised Principal Component Analysis (PCA). (**A**) Projection based on the total spectral regions from the ^1^H HRMAS NMR spectra (see Appendix A). (**B**) Projection based on the spectral regions corresponding to amino acids from the ^1^H HRMAS NMR spectra (Appendix A). (**C**) Distribution of amino acids within the same PC1–PC2 projection space. Red colored shapes related to I-CLM, blue ones to T-CLM.

**Figure 4 foods-14-02355-f004:**
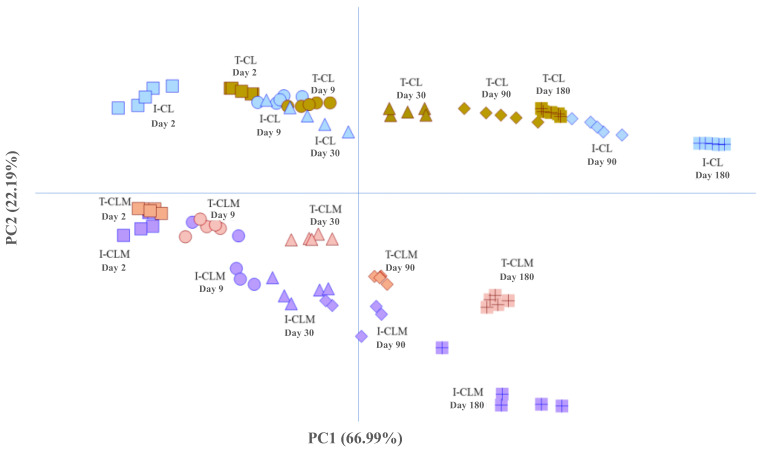
Distribution of pressed-curd cheese samples made from ewe’s milk using enzymatic coagulation—*Manchego* (CLM) and *Castellano* (CL)—produced through industrial (I) and traditional (T) methods at different ripening times (days), projected onto the first two principal components (PC1 and PC2) obtained from unsupervised Principal Component Analysis (PCA) using the spectral regions (buckets) of the ^1^H HRMAS NMR spectra corresponding to amino acids (Appendix A).

**Figure 5 foods-14-02355-f005:**
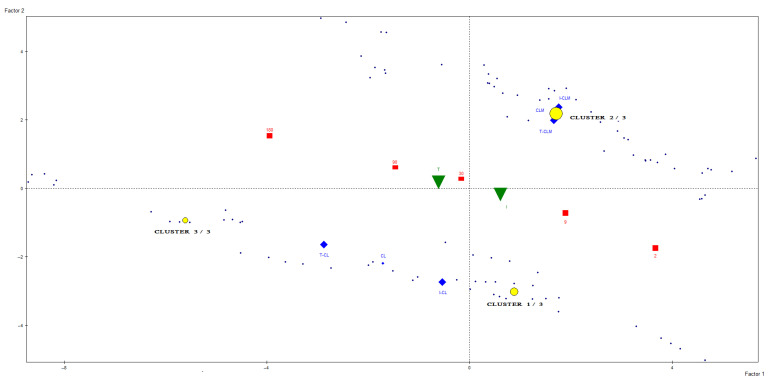
Distribution of *Manchego* (CLM) and *Castellano* (CL) ewe’s milk cheese samples, produced through industrial (I) and traditional (T) methods at different ripening times, along with the representation of centroids for the three clusters obtained from the spectral regions of the ^1^H HRMAS NMR spectra. The characteristics of the clusters are presented in Table 3.

**Table 1 foods-14-02355-t001:** Relative signal intensity (arbitrary units, AU) of butyric acid, caproleic acid, linoleic acid, conjugated linoleic acid (CLA), and glycerol from ^1^H HRMAS NMR spectra.

Fatty Acids *	Samples	Ripening Time (Days)																								
		2					9					30					90					180				
Butyric acid	I-CL	5.780	±	0.0250	a	α	5.690	±	0.0405	b	α	5.550	±	0.0301	c	α	5.490	±	0.0620	cd	α	5.400	±	0.0444	d	α
	T-CL	0.781	±	0.0111	c	β	0.801	±	0.0112	c	β	0.922	±	0.0103	b	β	1.320	±	0.0170	a	β	1.390	±	0.0166	a	β
	I-CLM	8.200	±	0.0539	ab	α	8.769	±	0.1330	a	α	8.796	±	0.0386	a	α	7.478	±	0.0915	b		7.573	±	0.0959	b	β
	T-CLM	6.968	±	0.0102	b	β	6.948	±	0.0014	b	β	6.698	±	0.0018	b	β	7.354	±	0.0345	ab		7.816	±	0.0033	a	α
Caproleic acid	I-CL	0.009	±	0.0015		β	0.009	±	0.0014		β	0.009	±	0.0002		β	0.008	±	0.0015		β	0.008	±	0.0013		β
	T-CL	0.010	±	0.0016		α	0.011	±	0.0012		α	0.011	±	0.0017		α	0.011	±	0.0004		α	0.012	±	0.0004		α
	I-CLM	0.016	±	0.0036	a	α	0.015	±	0.0012	a	α	0.014	±	0.0007	ab		0.014	±	0.0014	ab		0.013	±	0.0019	b	
	T-CLM	0.011	±	0.0016	b	β	0.012	±	0.0026	ab	β	0.013	±	0.0092	a		0.014	±	0.0063	a		0.014	±	0.0011	a	
Linoleic acid	I-CL	0.121	±	0.0028		α	0.128	±	0.0030		α	0.130	±	0.0019		α	0.131	±	0.0039		α	0.139	±	0.0017		α
	T-CL	0.027	±	0.0034	b	β	0.038	±	0.0021	b	β	0.049	±	0.0020	ab	β	0.063	±	0.0008	a	β	0.066	±	0.0043	a	β
	I-CLM	0.092	±	0.0049			0.104	±	0.0099			0.110	±	0.0041			0.117	±	0.0033			0.127	±	0.0125		
	T-CLM	0.089	±	0.0020			0.090	±	0.0029			0.091	±	0.0057			0.105	±	0.0020			0.114	±	0.0028		
Linolenic acid	I-CL	0.258	±	0.0024		β	0.261	±	0.0023		β	0.277	±	0.0017		β	0.286	±	0.0052		β	0.290	±	0.0012		β
	T-CL	0.291	±	0.0042		α	0.295	±	0.0013		α	0.298	±	0.0010		α	0.301	±	0.0018		α	0.309	±	0.0028		α
	I-CLM	0.295	±	0.0126		α	0.276	±	0.0254		α	0.272	±	0.0076			0.274	±	0.0172			0.289	±	0.0174		
	T-CLM	0.255	±	0.0052		β	0.247	±	0.0228		β	0.277	±	0.0042			0.268	±	0.0051			0.280	±	0.0028		
CLA	I-CL	0.009	±	0.0005	b	β	0.009	±	0.0004	b	β	0.009	±	0.0006	b	β	0.008	±	0.0006	a	β	0.009	±	0.0007	a	β
	T-CL	0.013	±	0.0004	b	α	0.010	±	0.0002	b	α	0.012	±	0.0006	ab	α	0.013	±	0.0003	a	α	0.014	±	0.0002	a	α
	I-CLM	0.013	±	0.0033	b		0.014	±	0.0016	b		0.014	±	0.0022	ab		0.015	±	0.0021	a		0.015	±	0.0006	a	
	T-CLM	0.013	±	0.0078	b		0.015	±	0.0051	ab		0.015	±	0.0044	ab		0.016	±	0.0014	a		0.016	±	0.0044	a	

* Relative signal intensity (AU) corresponding to the ratio of the NMR signal integral to the sum of all NMR signal integrals observed. Elaboration procedures, industrial (I) and traditional (T), according to Appendix A. Different letters a, b … within a row indicate statistically significant differences (*p* < 0.05) regarding RT. Different letters α, β … within a column for each compound indicate statistically significant differences (*p* < 0.05) regarding the manufacturing process (T vs. I).

**Table 2 foods-14-02355-t002:** Relative signal intensity (arbitrary units, AU) of the principal detected amino acids from ^1^H HRMAS NMR spectra.

Amino Acids *	Samples	Ripening Time (Days)																								
		2					9					30					90					180				
Histidine (His)	I-CL	0.205	±	0.060	b		0.199	±	0.045	b		0.285	±	0.037	b	β	0.612	±	0.090	a	β	0.747	±	0.012	a	β
	T-CL	0.215	±	0.022	b		0.196	±	0.124	b		1.539	±	0.092	a	α	1.355	±	0.029	a	α	1.505	±	0.022	a	α
	I-CLM	0.320	±	0.080	c	α	0.370	±	0.120	c		0.450	±	0.050	b		0.630	±	0.190	a	β	0.700	±	0.070	a	β
	T-CLM	0.260	±	0.010	c	β	0.360	±	0.100	bc		0.410	±	0.010	b		0.940	±	0.010	a	α	1.550	±	0.010	a	α
Phenylalanine (Phe)	I-CL	0.258	±	0.033	c		0.280	±	0.011	c		0.542	±	0.023	b	β	1.950	±	0.096	a	β	2.254	±	0.039	a	β
	T-CL	0.193	±	0.009	c		0.344	±	0.194	c		2.353	±	0.114	b	α	3.292	±	0.018	a	α	3.586	±	0.022	a	α
	I-CLM	1.250	±	0.120	c	α	1.810	±	0.780	bc		2.950	±	0.500	b	β	4.430	±	0.830	a		5.790	±	1.850	a	
	T-CLM	0.860	±	0.030	d	β	1.590	±	0.060	c		4.120	±	0.100	b	α	4.970	±	0.120	b		6.170	±	0.010	a	
Tryptophan (Trp)	I-CL	0.603	±	0.040	c		0.700	±	0.033	bc		0.852	±	0.041	b	β	1.450	±	0.001	a		1.564	±	0.148	a	
	T-CL	0.418	±	0.007	c		0.788	±	0.187	b		1.657	±	0.175	a	α	1.889	±	0.307	a		1.946	±	0.050	a	
	I-CLM	0.430	±	0.140	c		0.690	±	0.110	b	α	1.000	±	0.270	a		1.180	±	0.260	a		1.710	±	0.540	a	
	T-CLM	0.300	±	0.040	d		0.470	±	0.010	c	β	0.860	±	0.010	b		1.250	±	0.120	a		1.130	±	0.030	a	
Tyrosine (Tyr)	I-CL	0.323	±	0.048	c		0.395	±	0.016	c		0.511	±	0.022	b	β	1.039	±	0.078	a	β	1.257	±	0.031	a	β
	T-CL	0.243	±	0.033	d		0.484	±	0.070	c		1.551	±	0.137	b	α	2.225	±	0.030	a	α	2.551	±	0.041	a	α
	I-CLM	2.150	±	0.220	d		4.500	±	0.090	c	α	4.650	±	0.070	c	β	5.700	±	0.080	b	β	6.230	±	0.080	a	β
	T-CLM	2.010	±	0.050	e		3.670	±	0.030	d	β	5.870	±	0.060	c	α	6.810	±	0.010	b	α	9.000	±	0.010	a	α
Threonine (Thr)	I-CL	2.630	±	0.372	b		2.808	±	0.202	b		2.902	±	0.086	b	β	4.060	±	0.264	a	β	4.643	±	0.440	a	β
	T-CL	2.590	±	0.066	d		3.781	±	0.455	c		3.908	±	0.162	c	α	6.280	±	0.258	b	α	7.967	±	0.141	a	α
	I-CLM	1.660	±	0.690	c		2.730	±	0.880	bc	β	3.280	±	0.350	b	β	3.790	±	0.410	b	β	6.100	±	0.520	a	α
	T-CLM	2.280	±	0.060	d		3.220	±	0.080	c	α	4.270	±	0.220	b	α	5.130	±	0.080	a	α	5.630	±	0.060	a	β
Proline (Pro)	I-CL	1.160	±	0.306	b		1.197	±	0.116	b		1.116	±	0.131	b		1.660	±	0.486	a	β	1.876	±	0.220	a	β
	T-CL	0.853	±	0.073	c		1.466	±	0.368	b		1.947	±	0.228	b		3.103	±	0.332	a	α	3.518	±	0.245	a	α
	I-CLM	0.600	±	0.060	d		1.400	±	0.180	c	α	2.000	±	0.250	b	α	2.440	±	0.070	b	α	4.120	±	0.370	a	α
	T-CLM	0.470	±	0.120	c		1.070	±	0.180	b	β	1.570	±	0.330	b	β	2.180	±	0.150	a	β	2.660	±	0.110	a	β
Leucine (Leu)	I-CL	2.720	±	0.512	b		2.860	±	1.331	b		3.008	±	0.535	a		3.569	±	1.172	a		3.696	±	0.030	a	
	T-CL	2.674	±	1.258	b		2.861	±	0.218	b		3.647	±	0.325	a		4.102	±	1.200	a		4.296	±	0.167	a	
	I-CLM	7.060	±	0.834	c	α	9.190	±	1.890	b	α	10.900	±	0.633	ab	α	12.300	±	1.540	a	α	15.300	±	2.660	a	α
	T-CLM	5.340	±	0.254	d	β	6.020	±	0.029	c	β	6.180	±	0.029	c	β	8.260	±	0.143	b	β	9.530	±	0.051	a	β
Isoleucine (Ile)	I-CL	9.413	±	1.197	b		9.732	±	0.794	ab		10.562	±	0.168	a		11.130	±	1.921	a		11.481	±	0.145	a	
	T-CL	7.810	±	2.728	b		9.032	±	1.679	b		12.685	±	0.076	a		13.160	±	1.728	a		13.430	±	0.306	a	
	I-CLM	2.650	±	0.249	c		2.800	±	0.147	c		3.380	±	0.134	b	α	3.620	±	0.104	b	α	4.810	±	0.286	a	α
	T-CLM	2.420	±	0.064	d		2.660	±	0.005	c		3.090	±	0.064	b	β	3.290	±	0.004	b	β	4.380	±	0.164	a	β
Aspartic Acid (Asp)	I-CL	2.460	±	0.038	b		2.607	±	0.047	b		2.976	±	0.052	b	β	4.230	±	0.114	a	β	4.640	±	0.031	a	β
	T-CL	1.980	±	0.048	d		2.748	±	0.047	c		5.541	±	0.024	b	α	6.460	±	0.012	a	α	6.785	±	0.023	a	α
	I-CLM	1.542	±	0.610	c		2.310	±	0.490	c	α	2.888	±	0.460	bc		3.307	±	0.100	b		5.304	±	1.020	a	
	T-CLM	1.585	±	0.170	c		1.965	±	0.210	c	β	3.043	±	0.080	b		3.569	±	0.080	b		5.183	±	0.180	a	
Glutamine (Gln)	I-CL	5.530	±	0.026	b	α	5.708	±	0.072	b	α	5.934	±	0.044	b	β	7.020	±	0.160	a	β	7.347	±	0.020	a	β
	T-CL	3.620	±	0.090	c	β	4.646	±	0.002	c	β	7.936	±	0.008	b	α	8.780	±	0.033	a	α	9.187	±	0.058	a	α
	I-CLM	4.110	±	0.280	c		4.340	±	0.630	bc	β	5.780	±	0.920	b	β	6.170	±	0.170	b	β	8.920	±	0.110	a	β
	T-CLM	3.670	±	0.810	c		5.730	±	0.680	b	α	6.910	±	0.750	b	α	7.270	±	0.550	b	α	10.100	±	1.440	a	α
Glutamic Acid (Glu)	I-CL	1.477	±	0.023	c		1.567	±	0.057	bc		1.646	±	0.067	b	β	2.279	±	0.136	a		2.495	±	0.045	a	β
	T-CL	1.267	±	0.119	c		1.487	±	0.012	c		2.513	±	0.006	b	α	2.837	±	0.033	a		3.032	±	0.048	a	α
	I-CLM	1.240	±	0.017	c		1.410	±	0.086	c		1.740	±	0.110	b		1.840	±	0.023	b		2.580	±	0.056	a	β
	T-CLM	1.350	±	0.109	c		1.520	±	0.169	bc		1.840	±	0.134	b		2.050	±	0.173	b		3.040	±	0.363	a	α
Methionine (Met)	I-CL	3.667	±	0.116	b	β	3.902	±	0.035	b	β	3.868	±	0.052	b	β	4.661	±	0.067	a	β	4.554	±	0.204	a	β
	T-CL	5.314	±	0.256	b	α	4.546	±	0.195	b	α	9.326	±	0.021	a	α	9.939	±	0.021	a	α	9.978	±	0.098	a	α
	I-CLM	1.380	±	0.670	b		1.940	±	0.410	b		2.080	±	0.400	b		2.250	±	0.100	ab		3.020	±	0.440	a	β
	T-CLM	1.400	±	0.070	c		1.800	±	0.320	c		2.790	±	0.600	b		2.610	±	0.020	b		4.190	±	0.330	a	α
Valine (Val)	I-CL	2.228	±	0.103	b		2.339	±	0.192	b		2.516	±	0.099	b		3.194	±	0.216	a		3.352	±	0.065	a	
	T-CL	1.986	±	0.464	c		2.206	±	0.266	c		3.039	±	0.650	b		3.512	±	0.034	a		3.702	±	0.256	a	
	I-CLM	3.680	±	0.610	c	β	6.400	±	0.218	b	α	6.420	±	0.900	b	β	7.550	±	0.470	a	β	8.920	±	0.970	a	β
	T-CLM	4.540	±	0.160	c	α	4.530	±	0.120	c	β	9.000	±	0.030	b	α	9.400	±	0.240	b	α	11.500	±	0.150	a	α

* Relative signal intensity (arbitrary units, AU) corresponding to the ratio of the NMR signal integral to the sum of all NMR signal integrals observed. Elaboration procedures, industrial (I) and traditional (T), according to Appendix A. Different letters a, b … within a row indicate statistically significant differences (*p* < 0.05) regarding RT. Different letters α, β … within a column for each compound indicate statistically significant differences (*p* < 0.05) regarding the manufacturing process (T vs. I).

**Table 3 foods-14-02355-t003:** Defining parameters of the clusters obtained through the clustering analysis based on the spectral regions (buckets) of the ^1^H HRMAS NMR spectrum of cheeses from Castilla y León (CL) and Castilla–La Mancha (CLM) produced industrially (I) and traditionally (T) with different ripening times (2, 9, 30, 60, 180 days).

	Cluster 1				Cluster 2				Cluster 3			
Characteristic	Characteristic	Group Category (%)	*t*-Student	*p*-Value	Characteristic	Group Category (%)	*t*-Student	*p*-Value	Characteristic	Group Category (%)	*t*-Student	*p*-Value
**Geography**	CL	100	6.79	0.0001	CLM	100	11.28	0.0000	CL	100	5.22	0.0000
	CLM	0	−6.97	0.0000	CL	0	−11.28	0.0000	CLM	0	−5.22	0.0000
**Type**	I-CL	66.7	5.93	0.0001	T-CLM	50	6.1	0.0000	T-CL	75	5.18	0.0000
					I-CLM	50	6.1	0.0000				
**Ripening time (days)**	180	0	−3.43	0.0001					180	50	3.22	0.001
									2	0	−2.48	0.007
									9	0	−2.48	0.007
**Metabolite**	**Mean**	**SD**	***t*-Student**	***p*-Value**	**Mean**	**SD**	***t*-Student**	***p*-Value**	**Mean**	**SD**	***t*-Student**	***p*-Value**
Aspartic acid	0.0003	0.0001	−3.4549	0.0003	0.0003	0.0001	−2.4731	0.0067	0.0006	0.0001	7.0495	0.0000
Glutamic acid	0.0358	0.0010	6.5192	0.0000	0.0019	0.0006	−9.9411	0.0000	0.0357	0.0003	4.9577	0.0000
Asparagine	0.0004	0.0001	−2.4876	0.0064	0.0004	0.0001	−2.9566	0.0016	0.0006	0.0000	6.5457	0.0000
Phenylalanine	0.0002	0.0001	−3.9648	0.0000					0.0007	0.0001	7.1066	0.0000
Glutamine					0.0006	0.0002	−8.2071	0.0000	0.0027	0.0002	8.2419	0.0000
Histidine	0.0000	0.0000	−5.1925	0.0000					0.0001	0.0000	6.6514	0.0000
Isoleucine	0.0029	0.0003	−4.1080	0.0000					0.0039	0.0003	4.3854	0.0000
Leucine	0.0037	0.0008	−6.0491	0.0000	0.0090	0.0032	6.6859	0.0000				
Methionine					0.0002	0.0001	−8.0185	0.0000	0.0008	0.0001	7.6561	0.0000
Proline	0.0001	0.0000	−4.1270	0.0000					0.0003	0.0001	4.1025	0.0000
Tyrosine	0.0000	0.0000	−3.3130	0.0005	0.0001	0.0000	−3.9128	0.0000	0.0002	0.0000	8.6866	0.0000
Threonine	0.0003	0.0001	−4.1312	0.0000					0.0005	0.0002	4.6800	0.0000
Tryptophan	0.0001	0.0000	−3.6459	0.0001					0.0002	0.0000	6.6877	0.0000
Valine	0.0006	0.0001	−3.6326	0.0001	0.0007	0.0002	−2.5991	0.0047	0.0013	0.0002	7.4105	0.0000
Acetic acid	0.0015	0.0003	−7.0440	0.0000	0.0059	0.0017	8.2407	0.0000				
Citric acid	0.0006	0.0001	3.0169	0.0013	0.0004	0.0001	−7.5405	0.0000	0.0008	0.0001	5.9694	0.0000
Lactic acid	0.0042	0.0008	−3.1572	0.0008	0.0045	0.0008	−2.4370	0.0074	0.0065	0.0014	6.6632	0.0000
Ethanol	0.0078	0.0004	−3.2882	0.0005	0.0082	0.0005	3.0032	0.0013				
Choline	0.0002	0.0001	−3.1532	0.0008					0.0004	0.0001	4.9108	0.0000
Cholesterol	0.0013	0.0001	5.5490	0.0000	0.0006	0.0001	−9.4532	0.0000	0.0013	0.0001	5.4593	0.0000
Butyric acid	0.0093	0.0014	−6.2615	0.0000	0.0524	0.0138	8.9918	0.0000	0.0126	0.0007	−4.0664	0.0000
Caproleic acid	0.0001	0.0000	−2.7537	0.0029	0.0001	0.0000	5.1125	0.0000	0.0001	0.0000	−3.2360	0.0006
CLA	0.0001	0.0000	−5.4451	0.0000	0.0001	0.0000	7.7723	0.0000	0.0001	0.0000	−3.4772	0.0003
Linoleic acid					0.0010	0.0001	−6.2813	0.0000	0.0015	0.0001	7.2087	0.0000

## Data Availability

The original contributions presented in the study are included in the article/Appendix A, further inquiries can be directed to the corresponding author.

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
