# Peer review of "1H HRMAS NMR Metabolomics for the Characterization and Monitoring of Ripening in Pressed-Curd Ewe’s Milk Cheeses Produced Through Enzymatic Coagulation"

_foods, 2025, doi:10.3390/foods14132355_

Round 1
Reviewer 1 Report
Comments and Suggestions for Authors
This study describes the application of ¹H HRMAS NMR spectroscopy to two pressed sheep’s milk cheeses, with the aim of characterizing and distinguishing their geographical origin, production method, and degree of ripening. The goal is to develop a non-destructive technique particularly well suited to the analysis of complex food matrices and characterize the metabolome in order to definine of a specific fingerprint for the product.
I would like to thank the authors for their interesting and well-structured manuscript. The topic is relevant, and the work is generally well conducted and clearly presented.
Nevertheless, the manuscript needs some corrections and improvements before being published, detailed below:
Most of the suggestions and corrections have been made directly in the draft PDF file; the remaining comments are provided below:
- In Table S1, in the last section “Ripening”, for T-CL the temperature of chamber in natural condition is 10-12 T (°C) and RH (%) 75-80, but what does means natural condition? what keeps the temperature and humidity constant in ripening chambers?
- Please standardize the formatting of Table 2.
- I suggest adding the amino acid abbreviations (three letter codes) inside Table 2 under the full name of the single amino acid.
- In Figure 3 (C) you have mistakenly written Asn (Asparisian) instead of Asp (Aspartic acid), the same mistake is carried over into the legend of the Figure 3 (rows 448, 450, 451) and in Figure 4 (row 496).
- Please improve the resolution of the image in Figure 5, it is too blurry and the writing is not clear.
- Nei risultati dici che l'acido caproleico diminuisce rispetto alla media ma dalla Tabella S4 non risulta così (riga 517).
- La tabella S4 è frequentemente citata nella sezione Risultati ed è essenziale per seguire la discussione; Suggerisco di includerlo nel manoscritto principale.
- Nella tabella S3 l'acido lattico è sottolineato in modo errato, si prega di correggere.

Reviewer 2 Report
Comments and Suggestions for Authors
In this study, ¹H HRMAS NMR spectroscopy was employed to monitor the metabolomic evolution throughout the ripening period of two pressed-curd cheeses made from ewe's milk using enzymatic coagulation, produced in Spain. The findings obtained in this study may serve as a foundation for future research on this category of cheeses or similar dairy products. Overall, the manuscript is well organized and its presentation is good. There are still some questions in this manuscript remain to be fixed.
(1) line 91. the combination of 1H HRMAS NMR spectroscopy with multivariate chemomet-91 ric methods represents a highly effective approach. What are the chemometric methods available? What are the applications of these methods? Add references and examples.
(2) line 124. The ewe's milk cheeses (EMC), referred to as CLM and CL throughout this study, 124 were produced following the procedures. How to ensure full coverage of sampling? How to ensure the significance of this study.
(3) Figure 1. Add an explanation and unit for the horizontal axis.
(4) Table 2. The font should be consistent.
(5) Figure 2. A) should be A.
(6) line 189 The number of principal components (PC) employed for PCA was established as the minimum required to explain 95% of the total variance. However, the PC1 46.13 % and PC2 19.83 %<<95%. How to explain?
(7) lines 447-450, delete. The explanation for this PC is incorrect.
(8) Figure 3A No coordinate explanation.
(9) line 491 The equations of PC: delete.
(10) Figure 5. Can't see clearly at all.
